# Understand, Think, and Answer: Advancing Visual Reasoning with Large Multimodal Models

## Abstract

Large Multimodal Models (LMMs) have recently demonstrated remarkable visual understanding performance on both vision-language and vision-centric tasks. However, they often fall short in integrating advanced, task-specific capabilities for compositional reasoning, which hinders their progress toward truly competent general vision models. To address this, we present a unified visual reasoning mechanism that enables LMMs to solve complicated compositional problems by leveraging their intrinsic capabilities (e.g., grounding and visual understanding capabilities). Different from the previous shortcut learning mechanism, our approach introduces a human-like understanding-thinking-answering process, allowing the model to complete all steps in a single pass, forwarding without the need for multiple inferences or external tools. This design bridges the gap between foundational visual capabilities and general question answering, encouraging LMMs to generate faithful and traceable responses for complex visual reasoning. Meanwhile, we curate 334K visual instruction samples covering both general scenes and text-rich scenes and involving multiple foundational visual capabilities. Our trained model, Griffon-R, has the ability to end-to-end automatic understanding, self-thinking, and reasoning answers. Comprehensive experiments show that Griffon-R not only achieves advanced performance on complex visual reasoning benchmarks, including VSR and CLEVR, but also enhances multimodal capabilities across various benchmarks like MMBench and ScienceQA.

## 1 Introduction

Inspired by the success of Large Language Models like ChatGPT OpenAI (2023) and Gemini Team et al. (2023), the vision field has been seeking to equip these models with visual understanding capabilities, aiming to replicate similar achievements in visual tasks. Currently, Large Multimodal Models (LMMs) Liu et al. (2024b); Li et al. (2023b); Dai et al. (2023); Li et al. (2024b); Liu et al. adopt a paradigm in which images are encoded and projected into a textual embedding space, then combined with language input to generate responses via the LLM Chiang et al. (2023); Gao et al. (2023). Trained with millions of high-quality data, LMMs demonstrate advancing performance across various vision-language tasks, such as visual question answering (VQA) Goyal et al. (2019) and image captioning Chen et al. (2015), and become increasingly proficient in fine-grained visual tasks like visual grounding Plummer et al. (2017) and object detection Lin et al. (2014), even surpassing specialized vision expert models Ren et al. (2015); Kamath et al. (2021) in certain domains.

Despite significant progress across a wide range of tasks, LMMs still fall short in visual reasoning tasks. Existing open-source LMMs mainly follow the shortcut learning paradigmGeirhos et al. (2020) and are trained to directly generate the final answer based on the question. As shown in Fig. 1(a), LMMs do well in visual foundational tasks that follow the shortcut paradigm like object recognition(Q1) and visual grounding(Q2). Though this paradigm benefits the LMMs a lot, it also hinders them from inferring based on the foundational visual capabilities, which are crucial for tackling compositional and complicated visual reasoning tasks. As indicated in Fig. 1(a), when directly asking the LMMs about the relative position of the red and white balloons, they respond with the unfaithful but confident answer, *i.e.* hallucination Shukang et al. (2023).

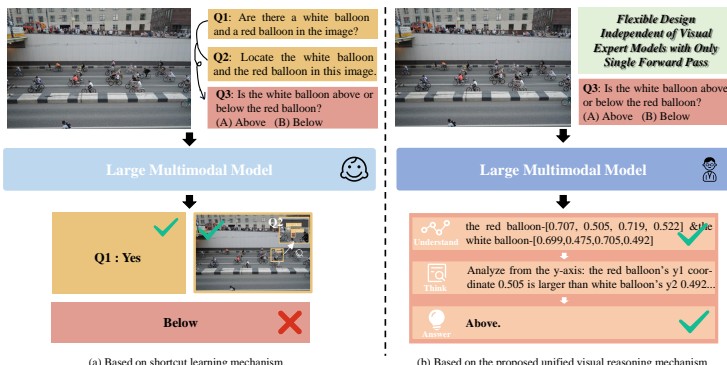

Figure 1: Enabled by the proposed unified mechanism, Griffon-R naturally connects the reasoning processes for locating each balloon with answering the spatial relationship question. It effectively analyzes their y-axis coordinates and provides the correct answer in a single pass.

To address these challenges, recent studies have shown that incorporating multimodal Chain-of-Thought (CoT) Shao et al. (2024); Li et al. (2024a); Mitra et al. (2024) to encourage LMMs to reason step by step can improve the reason quality. However, these CoT methods are specifically designed for target types of questions or domains and may require multiple times of forwarding. Other toolkit-based methods Qi et al. (2024); Jain et al. (2024); Wu & Xie (2024) enhance LMMs by directly supplementing details needed for specific tasks using the visual toolkit with the format of structured text or feature-based prompts. Though the visual toolkit can provide accurate information, calling it brings significant computational load and latency. Also, with the task complexity increasing, the parameters scale up exponentially.

In this paper, we propose a unified visual reasoning mechanism to enable LMMs to harness advanced intrinsic visual foundational capabilities for compositional visual reasoning in a single forward pass. Inspired by the human reasoning processAnderson (2005); McVee et al. (2005)—where individuals begin by contemplating how to answer a question, gather sufficient information from their environment, think with their experiences, and ultimately formulate an answer—our unified mechanism integrates this progressive "understand-think-answer" approach. Following this proposed process, the model first plans the necessary information acquisition for answering the question and generates structured instructions to autonomously gather relevant information, ensuring a thorough understanding. Considering the contextual understanding and extensive knowledge base of LLMs, rather than designing question-specific reasoning paths, the model is self-prompted to engage in contextual thinking after obtaining a comprehensive understanding. This design allows for greater flexibility across various question types. Finally, the model generates an answer, marking the conclusion of the visual reasoning process. This entire process operates without manual intervention or the need for external tools, achieving both efficiency and adaptability in a single forward pass. To implement this mechanism, we introduce a semi-automatic expert-supervised data engine and curate a dataset of 334K visual reasoning samples, encompassing both natural and textual scenes. This dataset is annotated progressively by employing AITeam (2024); Wang et al. (2024) and human experts in a streamlined pipeline. Ultimately, we train the Griffon-R model using this curated data to achieve the unified mechanism.

To validate our design, we conduct comprehensive experiments with Griffon-R across a range of visual reasoning and multimodal benchmarks. The results demonstrate that empowered by our design mechanism, Griffon-R achieves advancing performance on complex visual reasoning tasks VSRLiu et al. (2023a) and CLEVRJohnson et al. (2017) and surpasses advanced LMMs on multimodal tasks including MMbench Liu et al. (2023c), ScienceQA Lu et al. (2022), *etc.*, highlighting its enhanced general capabilities. Our key contributions are as follows:

- We propose a unified visual reasoning mechanism inspired by the human reasoning process, enabling LMMs to handle diverse compositional tasks by leveraging advanced capabilities through an "understand-think-answer" process in a single forward pass.

- We curate 334K multi-scene visual reasoning data by the introduced semi-automatic expert-supervised data engine and further present Griffon-R, a general LMM that is skilled in solving complicated compositional problems.

- We conduct extensive experiments on a wide range of visual reasoning and multimodal benchmarks. Griffon-R achieves advancing performance in compositional VSR and TallyQA, while further boosting performance on the multimodal benchmark including MMBench, ScienceQA, *etc.*.

## 2 RELATED WORKS

### 2.1 MULTIMODAL CHAIN OF THOUGHT

The CoT approach Wei et al. (2022); Kojima et al. (2022); Zhang et al. (2022); Yao et al. (2024); Besta et al. (2024) is a series of prompting techniques that improve the ability of LLMs to solve complex reasoning tasks. With the rise of LMMs, these methods have gradually been incorporated into the multimodal domain to enhance model performance on complex reasoning tasks. ScienceQA Lu et al. (2022) pioneeringly proposes Multimodal CoT to combine image captioning for reasoning, thus enabling models to handle complex question-answering tasks in science. Later works further decompose complex visual reasoning tasks into sequential steps, incorporating diverse prompts such as bounding boxes Chen & et al (2023); Li et al. (2024a); Shao et al. (2024), textual descriptions Zheng et al. (2023), and scene graphsMitra et al. (2024), allowing models to reduce intuitive errors by following a structured reasoning path. However, these multimodal CoT methods usually design specific paths for considered tasks and thereby limiting the model's ability to generalize across diverse question types and visual tasks. In this way, methods like Visual CoTShao et al. (2024) tend to provide rough prompts when questions are not included in these designed patterns. Also, some of themShao et al. (2024); Li et al. (2024a) re-forward regions during inference to enhance region understanding. In contrast, our approach keeps the streamlined model structure and inference process with a single forward pass, which is more efficient. Meanwhile, our mechanism analyzes the question to automatically leverage intrinsic capabilities to understand the image precisely, facilitating a more flexible pattern for better generalization.

### 2.2 TOOLKIT-BASED VISUAL REASONING

Unlike multimodal CoT methods that convert various prompts into text for guidance, visual-tool-using approachesLei et al. (2024); Yang et al. (2023); Jain et al. (2024); Qi et al. (2024); Wu & Xie (2024); Hu et al. (2024); Gupta & Kembhavi (2023); Surís et al. (2023) directly input information in different formats or modalities into the model, encouraging comprehensive understanding and reasoning. Methods like SoM Yang et al. (2023) and Scaffolding Lei et al. (2024) initially incorporate additional structured information within images, using these as anchors to prompt GPT-4V in visual reasoning. In contrast to these methods specifically designed for GPT-4V, VcoderJain et al. (2024) takes a different approach by projecting depth and segmentation maps into the text embedding space, thereby improving reasoning accuracy by enhancing model comprehension. These methods typically rely on one or two fixed visual tools, limiting the richness of the information provided. Consequently, other approaches Qi et al. (2024); Wu & Xie (2024); Hu et al. (2024); Gupta & Kembhavi (2023); Surís et al. (2023) have been developed to construct visual toolkits specifically for visual reasoning tasks in LMMs, allowing LLMs to generate executable program calls to specialist visual models based on the question. Building on this, CogCom Qi et al. (2024) further integrates certain visual manipulations with the model's internal capabilities, enabling it to generate and execute an operation chain to progressively complete reasoning tasks. In comparison to these methods, our method leverages intrinsic capabilities and supports knowledge sharing across contexts, minimizing the risk of errors from isolated task execution. Additionally, with an end-to-end manner, our approach reduces the optimization difficulty and the latency.

## 3 METHODOLOGY

In this section, we start with our designed novel unified visual reasoning mechanism, specifically the "Understand-Think-Answer" process which bridges the intrinsic visual foundational capabilities and

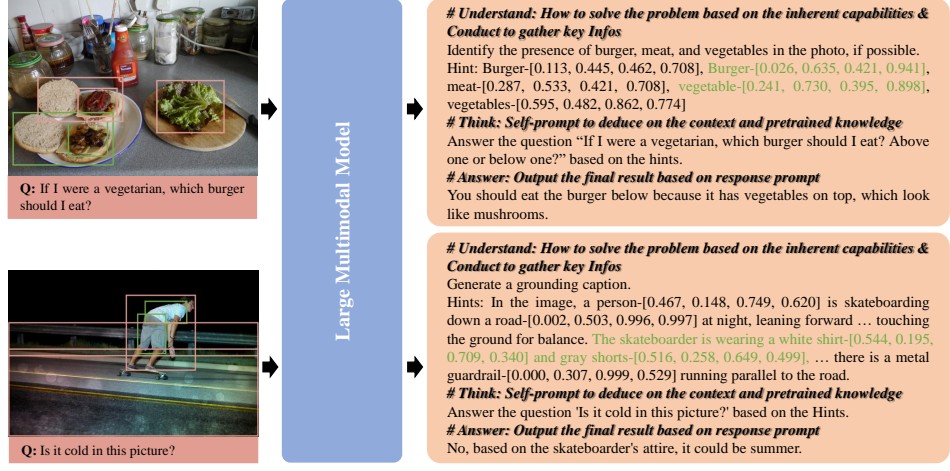

Figure 2: Detailed illustration of the unified visual reasoning mechanism with the "Understand-Think-Answer" process. The key information related to the answer is highlighted or visualized in green. We illustrate the details of the designed process in bold, which will not be generated or trained.

VQA answering to allow the LMMs to reason accurately in a single pass in Sec. 3.2. Then, we detail the semi-automatic expert-supervised data engine on how it curates the 334K high-quality visual reasoning data aligned with the mechanism in Sec. 3.3. Finally, we present the Griffon-R, an visual reasoning improved LMM built on the proposed unified mechanism and data.

## 3.1 PRELIMINARY

Current LMMs typically adopt an auto-regressive approach to generate response via next token prediction. Specifically, for an LMM $M$ with parameters $\theta$, given an input image $I$ and user instruction $Q$, the model maximizes the sequence probability to output the response sequence $X$:

$$p(X) = \prod_i p(x_i|I, Q, X_{<i}) \tag{1}$$

where $X_{<i}$ represents the sequence before the current prediction $x_i$ in the output sequence. Current LMMs employ a shortcut learning paradigm, training the model to directly generate the final answer $X_{ans}$. When simplifying the probability calculation process, the process can be denoted as:

$$X_{ans} = M_\theta(I, Q) \tag{2}$$

Though it effectively solves straightforward question-answering tasks, it struggles with more challenging compositional visual reasoning tasks, as illustrated in Fig. 1. Therefore, we propose the followed unified visual reasoning paradigm that advances the model to perform compositional reasoning leveraging intrinsic capabilities in a single forward pass through "understand-think-answer" process.

## 3.2 UNIFIED VISUAL REASONING MECHANISM

Current LLMsTeam et al. (2024); Team (2024) trained on billions of data have acquired rich knowledge and experience, while LMMs through image-text alignmentLi et al. (2023b); Radford et al. (2021) and instruction tuningDai et al. (2023); Liu et al. (2024b) further enhance their ability to perceive and interpret information by themselves. However, they still struggle to imitate how an educated person answers compositional questions by problem analyzing with past experience, relevant information gathering, and reaching a conclusion from thinkingAnderson (2005); McVee et al. (2005). Therefore, we propose the unified visual reasoning paradigm to imitate this approach to enable LMMs to sequentially and continuously perform the "understand-think-answer" process, accurately arriving at the answer in a single pass without introducing any toolkit.

**Understand.** Understanding is a key step that directly impacts the accuracy of the response. Unlike direct answering, the models first analyze the question and image to determine how to approach the

problem and which intrinsic abilities to use to extract relevant information. Based on this analysis, the model employs the appropriate capabilities to gather key information, achieving a thorough understanding of the image in relation to the question. As shown in Fig. 2, we combine question analysis with intrinsic capability-based information retrieval planning and generate instructions for acquiring the required information. Then, they automatically guide the model to gather relevant visual cues. For the above sample in Fig. 2, the model identifies that solving the question requires knowing the location of the burger and distinguishing which items contain meat or vegetables, as indicated in the instruction by the terms "burger," "meat," and "vegetables". The "Indentify the presense" process involves applying these capabilities to gather the necessary information. This understanding process leverages common capabilities of LMMs, including caption, grounded caption, visual grounding, text recognition, *etc.*, instead of relying on any fixed capability like scene analysisMitra et al. (2024) or RECShao et al. (2024). Moreover, when no relevant information is found, the model outputs none instead of providing vague cluesShao et al. (2024); Wu & Xie (2024), avoiding potential misleading.

**Think & Answer.** Instead of designing a specific reasoning path, we adopt a self-prompt approach. Previous studies show current modelsOpenAI (2024) can effectively respond to questions based on contextual cues. Additionally, modern LMMsYou & et al (2024); Yuan et al. (2024) excel at understanding coordinates-format object references in the context without needing additional forward passesLi et al. (2024a); Shao et al. (2024) for object perception. After achieving a deep understanding, we allow the model to generate the instruction to encourage the model to engage in self-thinking based on the visual cues, as indicated in Fig. 2. Ultimately, the model generates the final reasoning output according to the response template in the user's instruction. By now, the model performs question-customized deep understanding of the image in a single forward pass, engages in self-prompted thinking based on the generated cues, and outputs the final answer. This process can be summarized as:

$$p([X_U, X_T, X_{ans}]) = \prod_i p(x_i | I, Q, X_{<i})$$

$$X_{ans} = M_\theta(I, Q, X_U, X_T)$$

where $X_U$ denotes the context generated during the understanding and $X_T$ denotes the context generated during the thinking. The sequence inside the square brackets [] is generated with a left-to-right order of generation. Compared to shortcut learning-based methods, our unified visual reasoning mechanism integrates intrinsic capabilities to provide more accurate solutions to combinatorial visual reasoning tasks. In contrast to other CoT and toolkit-based approaches, it offers both flexibility and efficiency, which eliminates the need for multiple forward passes.

### 3.3 EXPERT-SUPERVISED DATA ENGINE

Beyond the design of the mechanism, training LMMs for accurate visual reasoning remains a significant challenge due to data limitations. To address this challenge and support the implementation of our designed visual reasoning mechanism, we introduce a semi-automatic expert-supervised annotation engine in this section, which is designed to generate high-quality visual reasoning data with visual cue annotations following our mechanism. As shown in Fig. 3, this semi-automated approach, supported by experts, enables us to generate high-quality data that enhances the model's visual reasoning capabilities through the unified reasoning mechanism.

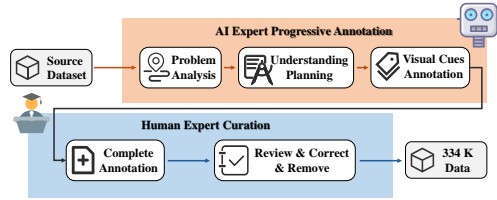

Figure 3: Illustration of the semi-automatic expert-supervised data generation engine.

**Progressive Annotation with AI Expert.** Due to the advanced question-answering capabilities and vast knowledge of current LMMs trained with billions of tokens, we leverage state-of-the-art LMMs to assist with annotation based on our visual reasoning mechanism. We select the Qwen2-VL-72B modelWang et al. (2024) from the latest open-source models as our AI Expert, due to its strong task performance. For each sample, the AI Expert is responsible for designing the understanding process for solving problems and annotating tasks it excels in. First, we prompt the AI Expert to analyze the question and image to determine how to solve the problem and identify the necessary information. Then, we further instruct the expert to outlining the understanding process with specific tasks to gather

key information based on the common intrinsic capabilities of LMMs. Finally, for tasks that the AI Expert specializes in, we directly use the model to generate the corresponding annotations, such as caption, text recognition, *etc*. The instructions used in this process are described in the Appendix A.

**Curation with Human Expert.** Although the AI Expert is skilled in analyzing and annotating tasks related to the question, its capabilities are limited in complex visual reasoning and tasks, such as visual grounding involving multiple objects. Previous worksChen et al. (2023); Yuan et al. (2024) have also highlighted that human-involved high-quality annotations form the foundation for scaling data size in subsequent phases. Therefore, after the AI Expert's annotation, we first employ human experts to complete tasks that the AI Expert is not proficient at, primarily visual grounding in scenes with multiple objects or partial text. Then, human experts further review the annotations for quality, including evaluating whether the understanding process is sound, ensuring the accuracy of the task annotations, and eliminating overly simplistic questions.

**Data Description.** Leveraging our proposed semi-automatic expert-supervised data generation pipeline, we curate 334K image-question pairs from multi-task instruction-following data across multiple levels from public datasets. We mainly focus on the general scene data and text-rich scene data, which covers compositional reasoning problems. The data annotations are then completed through the above semi-automatic expert-supervised annotation process, which incorporates the AI expert progressive annotation stage and human expert curation stages. We provide more information about the source data in the Appendix A.

### 3.4 GRIFFON-R

Leveraging the proposed unified visual reasoning mechanism and the curated 334K data, we further develop the Griffon-R model, an LMM proficient at both compositional visual reasoning tasks and straightforward QA tasks. Benefiting from our design, Griffon-R maintains a streamlined architecture without the need for additional perception structures. Also, Griffon-R generates responses only through the next token prediction, rather than relying on module calling or multiple forward passes. We detail the construction process, including the architecture and training pipeline, which may also serve as a guideline to facilitate the implementation of the unified visual reasoning mechanism.

**Structure.** Griffon-R adopts the advanced single-branch high-resolution structure proposed in Griffon v2 Zhan et al. (2024). Compared with other LMMs, this structure is proved to be better at fine-grained object localization, which is important for most of visual reasoning tasks. It consists of three core components: a high-resolution visual encoder, a vision-language connector, and an LLM. The high-resolution visual encoder processes image inputs up to 1K resolution without partitioning, with the connector projecting and compressing the tokens while retaining the performance.

**Training Pipeline.** We follow the common practice Li et al. (2024a); Shao et al. (2024) to train the Griffon-R model. Specifically, after the basic pertaining stage, we combine the curated visual reasoning data following our mechanism with VQA and instruction data to fine-tune the model with the whole model updated. Differently, we only use the cross-entropy loss to supervise the training without introducing task-specific lossesWu & Xie (2024). We detail the training data and setting for each stage in the Appendix B.1.

## 4 EXPERIMENTS

### 4.1 IMPLEMENTATION DETAILS

We follow the Griffon v2 Zhan et al. (2024) to set the resolution to 1022, randomly initialize a down-sampling projector implemented as a 3×3 convolution (stride 2, padding 1).Then, we utilize CLIP-ViT-L / 14-336 Radford et al. (2021) to initialize the visual encoder and further interpolate to the defined resolution. To ensure that our framework learns visual language reasoning capabilities from the scratch - without sacrificing generality - we selected Griffon-G-9B architectural paradigm and use Gemma9B Team et al. (2024) to initialize the LLM. For the training, we utilize the AdamW optimizer Loshchilov & Hutter (2017), setting the learning rate to 1e-3 for the first stage and 2e-5 for stage 2 and stage 3. We train each stage for 1 epoch with a batch size of 256. All training was performed on eight NVIDIA H100 GPUs.

Table 1: Evaluation results on visual reasoning benchmarks. CLEVR focuses on generated structured scenes, while the others evaluate models in the natural scenes. We use the Spatial subset of V-Star here due to the reasoning step included, while the attribute subset primarily evaluates models' attribute perception abilities in high-resolution scenes.

| Methods | VSR | CLEVR | GQA | V-Star$_{Spat.}$ | TallyQA | |
| | | | | | Simple | Complex |
|---|---|---|---|---|---|---|
| Large Multimodal Models | | | | | | |
| BLIP-2-7B Li et al. (2023b) | 50.9 | - | 44.7 | 53.9 | - | - |
| InstructBLIP-7B Dai et al. (2023) | 52.1 | - | 48.3 | 47.4 | 74.3 | 48.7 |
| LLaVA-1.5-7B Liu et al. (2023b) | 64.2 | 43.7 | 62.0 | 53.9 | 75.9 | 63.8 |
| LLaVA-1.5-13B Liu et al. (2023b) | 70.4 | 55.8 | 63.3 | 55.3 | 76.9 | 65.4 |
| Monkey-7B Li et al. (2024b) | 62.9 | 46.3 | 60.7 | 53.9 | 80.9 | 63.0 |
| DeepSeek-VL-7B Lu et al. (2024) | 67.5 | 48.8 | 61.31 | 40.3 | 79.5 | 62.1 |
| LLaVA-Next-7B Liu et al. (2024a) | 63.8 | 51.9 | 64.2 | 63.2 | 80.3 | 66.5 |
| Ferret v2-7B You et al. (2024) | - | - | 64.7 | - | - | - |
| Large Multimodal Models with MCoTs or Toolkit-Based Enhancement | | | | | | |
| VolCano-7B Li et al. (2024a) | 67.2 | 56.2 | 64.4 | 50.3 | 73.5 | 55.2 |
| CogCom-17B Qi et al. (2024) | - | - | **71.7** | - | 84.0 | 70.1 |
| SEAL-7B Wu & Xie (2024) | 48.5 | - | - | 76.3 | 51.9 | 20.6 |
| VPD-5B Hu et al. (2024) | - | - | 61.3 | - | 83.1 | **70.9** |
| VisualCoT-7B Shao et al. (2024) | 61.4 | 55.5 | 63.1 | 50.3 | 82.4 | 60.2 |
| VisualCoT-13B Shao et al. (2024) | - | 55.8 | 63.3 | 54.9 | 83.1 | 70.3 |
| Large Multimodal Models Leveraging Intrinsic Capabilities | | | | | | |
| Griffon-R-9B | **70.9** | **63.7** | 65.1 | **77.6** | **84.4** | 70.4 |

## 4.2 EVALUATION ON COMPOSITIONAL VISUAL REASONING

As shown in Tab. 1, Griffon-R achieves outstanding performance across these visual reasoning benchmarks. It reaches advancing levels in both generated scenes task CLEVR and natural scenes including VSR, V-Star$_{Spat.}$, and TallyQA$_{Simple}$, outperforming advanced visual reasoning models like Visual CoT and VPD. Specifically, Griffon-R achieves 63.7% accuracy on CLEVR and 70.9% accuracy on the VSR, surpassing the second-place models by a large margin. Such remarkable performance demonstrates the strong visual reasoning capabilities of Griffon-R across multiple tasks and scenarios. Notably, it also surpasses the latest advanced LMMs LLaVA-Next and Ferret v2. On the high-resolution, small-object compositional reasoning benchmark V-Star whose scenes are quite challenging for LMMs, Griffon-R outperforms multiple methods based on CoT and visual search techniques. These achievements across diverse task focus further highlight Griffon-R's robust visual reasoning capabilities and validate the effectiveness of our mechanism and data design.

## 4.3 EVALUATION ON MULTIMODAL BENCHMARKS

Empowered by the unified visual reasoning mechanism, Griffon-R leverages the model's inherent capabilities for visual reasoning tasks. Meanwhile, Griffon-R with streamlined structure and joint optimization with straightforward QA data can also handle these widely applied multimodal benchmarks. Therefore, we also evaluate Griffon-R on multimodal benchmarks and various VQA tasks to verify its comprehensive capabilities. As shown in Tab. 2, Griffon-R outperforms representative LMMs as well as advanced visual-reasoning-enhanced LMMs. Griffon-R achieves a score of 79.0 on the comprehensive MMBench and excels on ScienceQA, which focuses on scientific reasoning. Also, in real scenarios, Griffon-R is more proficient at challenging tasks like SEED and LLaVA-in-the-wild benchmark. In text-based scenes, Griffon-R further surpasses the program-driven VPD model, achieving 72.4% accuracy on TextVQA. These evaluations demonstrate that Griffon-R not only precisely handles complex compositional reasoning tasks but also performs well in general visual question answering.

Table 2: Evaluation results on multimodal benchmarks. Compared with visual reasoning benchmarks, these benchmarks focus more on visual understanding with common sense and knowledge, yet incorporating simple reasoning.

| Methods | MMB | ScienceQA | TextVQA | SEED-Img | LLaVA-W | POPE |
|---|---|---|---|---|---|---|
| Large Multimodal Models | | | | | | |
| InstructBLIP-7B Dai et al. (2023) | 36.0 | - | 50.1 | 58.8 | 60.9 | 72.1 |
| LLaVA-1.5-13B Liu et al. (2023b) | 67.7 | 71.6 | 61.3 | 68.2 | 72.5 | 85.9 |
| QwenVL-Chat-7B Bai & et al (2023) | 60.6 | 68.2 | 61.5 | 65.4 | - | 84.7 |
| Monkey-7B Li et al. (2024b) | 61.9 | 69.4 | 67.6 | 67.6 | 53.9 | 82.6 |
| DeepSeek-VL-7B Lu et al. (2024) | 71.3 | - | 63.7 | 70.4 | 21.6 | 85.8 |
| LLaVA-NeXT-7B Liu et al. (2024a) | 69.0 | 73.2 | 64.9 | 70.2 | 72.3 | 86.4 |
| Ferret v2-13B You et al. (2024) | - | - | 62.2 | 61.7 | 69.9 | 88.1 |
| Large Multimodal Models with MCoTs or Toolkit-Based Enhancement | | | | | | |
| VolCano-7B Li et al. (2024a) | 68.1 | 38.3 | 57.4 | 64.5 | 56.5 | 86.5 |
| SEAL-7B Wu & Xie (2024) | 33.1 | - | - | 41.7 | 59.1 | 82.4 |
| CCoT-13B Mitra et al. (2024) | 70.7 | 69.7 | - | 69.7 | 74.9 | - |
| VPD-5B Hu et al. (2024) | 69.0 | 83.1 | 70.9 | - | - | 88.6 |
| CogCom-17B Qi et al. (2024) | - | 84.0 | - | - | - | 87.8 |
| VisualCoT-7B Shao et al. (2024) | 67.3 | 68.3 | 61.0 | - | 49.7 | 86.5 |
| VisualCoT-13B Shao et al. (2024) | 67.4 | 73.6 | 62.3 | - | 57.7 | 83.3 |
| Large Multimodal Models Leveraging Intrinsic Capabilities | | | | | | |
| Griffon-R-9B | **79.0** | **87.0** | **72.4** | **73.8** | **76.2** | **89.3** |

## 4.4 ABLATION STUDIES

**Discussion on Understanding Quality.** As we have demonstrated in Sec. 3.2, understanding plays an important role in the whole mechanism for accurate visual reasoning. To quantitatively indicate the understanding quality, we choose the performance of Referring Expression Comprehension (REC) Yu et al. (2016); Nagaraja et al. (2016) as the metric, which incorporates both visual localization and attribute perception that are commonly used intrinsic capabilities during understanding for lots of reasoning tasks. As shown in Tab. 3, our model outperforms the SOTA method Ferret v2 in this task and also visual reasoning methods CogCom and Visual CoT. The result verifies that our model can

Table 3: Ablation on understanding quality. We report the average scores on RefCOCO/+/g.

| Methods | RefCOCO | RefCOCO+ | RefCOCOg |
|---|---|---|---|
| Expert Models | | | |
| UNINEXT | 92.80 | 84.87 | 89.05 |
| MDETR | 85.93 | 78.07 | 81.25 |
| G-DINO-L | 90.67 | 82.57 | 86.55 |
| Large Multimodal Models | | | |
| VistaLLM-7B | 87.53 | 82.50 | 84.00 |
| Ferret v2-7B | 92.07 | 86.50 | 89.35 |
| CogCom-17B | 92.03 | **87.93** | **89.90** |
| Visual CoT-7B | 91.20 | 86.93 | 88.40 |
| Griffon-R-9B | **92.83** | 87.83 | 89.75 |

achieve an accurate understanding with intrinsic capabilities, and it further facilitates the precise visual reasoning of Griffon-R.

**Discussion on the unified visual reasoning mechanism.** To validate the effectiveness of our unified visual reasoning mechanism, we compare its accuracy and inference time to the toolkit-based SEAL-7B method.We evaluate with the V-Star$_{Spat.}$ benchmarks and static the average inference time per-sample. As shown in Tab. 4, in the compositional reasoning task of V-Star, the designed mechanism outperforms the toolkit-based pipeline by 1.3 points and is 13x faster in average inference time. These results justify that our mechanism not only boost visual reasoning capabilities without relying on visual expert modules but also delivers high efficinet inference.

**Discussion on training with the curated data annotations.** In this section, we validate the necessity of training with curated annotations for our mechanism. The first line (baseline) and the second line results are trained on the same data with Griffon-R using the raw annotations. The second line additionally follows our mechanism but requires the model to answer step-by-step. We evaluate these methods on the V-Star$_{Spat.}$ complex visual reasoning benchmark and the POPE object existence

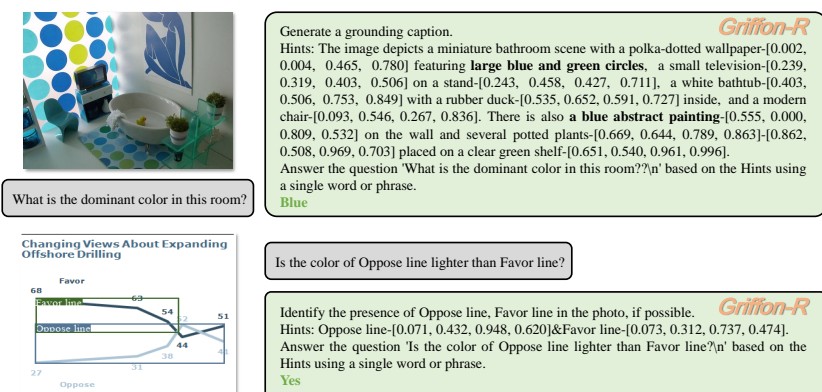

Figure 4: Visualization of Griffon-R's reasoning results. Correct answers are highlighted in bold green, and the relevant information within the long text leading to the answer is bolded.

Table 4: Ablation study on the unified visual reasoning mechanism and toolkit-based mechanism.

| Method | V-Star$_{Spat.}$ | Time/Sample |
|---|---|---|
| Unified(ours) | 77.6 | 0.336s |
| Toolkit-Based | 76.3 | 4.586s |

Table 5: Ablation study on the training and annotated data. UVRM denotes our mechanism.

| UVRM | 334k Ann. | V-Star$_{Spat.}$ | POPE |
|---|---|---|---|
| | | 75.0 | 89.1 |
| $\checkmark$ | | 75.0 | 89.4 (+0.3) |
| $\checkmark$ | $\checkmark$ | 77.6 (+2.6) | 89.3 (+0.2) |

hallucination benchmark. As shown in Tab. 5, simply providing visual cues (using UVRM) improves performance on tasks like POPE, which do not require compositional reasoning, by compensating for insufficient understanding. However, to achieve accurate visual reasoning, the model requires training on carefully curated datasets that enable it to understand the image, identify relevant cues, and reason based on different-pattern cues comprehension. This comparison emphasizes the effectiveness of our data-supported and unified visual reasoning mechanism design.

### 4.5 VISUALIZATION RESULTS

We display the qualitative performance of Griffon-R through Fig. 4. The results highlight Griffon-R effectively handles compositional visual reasoning across various scenes by thoroughly understanding the problem and visual cues, then performing context-based reasoning to generate the final answers.

## 5 CONCLUSION

In this paper, we present the unified visual reasoning mechanism that empowers LMMs to tackle compositional reasoning tasks end-to-end, without external expert models or toolkits. This unified mechanism introduces a human-like understand-think-answer process to reason based on individual questions flexibly instead of utilizing fixed paths and completes all steps in a single pass forward without multiple inferences. Moreover, we design a semi-automatic expert-supervised data generation engine to produce high-quality visual reasoning data corresponding to the design mechanism. We collect public data related to visual reasoning and re-annotate them with our designed data generation engine, and curate 334K visual instruction samples. Based on the curated data and designed mechanism, we present Griffon-R. Griffon-R achieves advancing results on visual reasoning benchmarks, including VSR and CLEVR, and also demonstrates comprehensive multimodal capabilities. We hope our attempts at a unified visual reasoning mechanism will facilitate the deep exploration in this field to achieve more general LMMs. In the Appendix, we provide a detailed discussion of our method's limitations and its broader impact. In future work, we plan to explore additional reasoning paradigms and more diverse data to extend the applicability of our approach.

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
