# A DETAILS OF EXPERT-SUPERVISED DATA ENGINE

In this section, we illustrate the details of data curation process with our proposed semi-automatic expert supervised data engine of Sec. 3.3. We start with the raw data collection, and then the instructions for the progressive annotation with AI expert.

## A.1 RAW DATA CURATION

As described in Sec. 3.3, we first collect the widely used diverse types of data that related to visual reasoning following the previous practice Wu & Xie (2024); Li et al. (2024a); Shao et al. (2024) and curate a total of 334K visual-reasoning-oriented data using the proposed Expert-Supervised Data Generation Engine. These data are composed of two main parts: VQA-based data and caption data tailored to visual reasoning tasks.

Table 6: Annotation sources of the 334K visual-reasoning data.

| Type | Num. | Source |
|---|---|---|
| VQA | 207K | GQA Hudson & Manning (2019), VAW Pham et al. (2021), VizWiz Bigham et al. (2010), ChartQA Masry et al. (2022), DUE_Benchmark Borchmann et al. (2021), TextVQA Singh et al. (2019) |
| Instruction | 76K | LLaVA Liu et al. (2024b), ALLaVA Chen et al. (2024), LVIS-Instruct4V Wang et al. (2023b) |
| Caption | 51K | ShareGPT-4V Chen et al. (2023) |

**VQA-Based Data.** The VQA-based data source from the general VQA datasets, instruction-following data, and text-oriented VQA datasets. We extract visual reasoning data based on two criteria: (1) Yes/No Answers: We select questions with "yes" or "no" answers, as these often involve tasks like comparisons or attribute judgments (e.g., color differentiation). (2) Reasoning Keywords: Using the relationship and attribute keywords appearing in GQA Hudson & Manning (2019), VAW Pham et al. (2021), and DUE_Benchmark Borchmann et al. (2021), which we extract with the raw keywords in the annotations or with spaCy Honnibal et al. (2020) (*eg.*, "left", "right", "over", "red", and "plastic"), we identify Q&A pairs that included these keywords. As summarized in Tab. 6, this process yielded 207K data from VQA annotations and 76K entries from instruction-based dialogues. For VAW data, we specifically utilize images aligned with GQA to generate more diverse visual reasoning questions. After acquiring these visual-reasoning-oriented VQAs, we follow the proposed data generation method to annotate these data.

Table 7: Instruction templates for the AI expert.

| Type | Template |
|---|---|
| I.1 | For the question:'{question}', identify and focus on the specific physical objects or entities relevant to answering it. Directly output the names of these objects or short descriptive phrases with key attributes. If the object is mentioned in the question, use the exact word from the question. If the object is not explicitly mentioned, describe it concisely using your own terms. |
| I.2 | For the question:'{question}', based on the context, identify the task used to gather the key entities information like [*TASKs*], if the question is too abstract, respond with "Global Understanding". |

**Visual-Reasoning-Oriented Caption Data.** Caption data provide global context and are essential for the in-depth understanding of images. To make them suitable for visual reasoning, we apply the proposed data generation method, focusing first on identifying the key objects in the image that are critical for reasoning tasks. The annotation process involved checking whether these key objects are included, fixing them if missing, and manually labeling their bounding boxes. We utilized the detailed descriptions generated by ShareGPT-4V Chen et al. (2023). After the above processing, we obtain the final expert-supervised visual-reasoning-oriented data, as listed in Tab. 6.

## A.2 AI EXPERT ANNOTATION DETAILS

At the beginning of data generation process, we first utilize the AI expert Qwen2-VL-72B Wang et al. (2024) to identify how to solve this problem with key objects or information listed (I.1) and plan how to gather these key information with intrinsic capabilities (I.2). We list the instructions to prompt the expert to finish these two tasks in Tab. 7 respectively. Also, as we have mentioned in Sec. 3.3, for the task annotation process that the AI expert is skilled at, we use the AI Expert to finish the annotation first. We mainly apply this strategy in tasks that require a global understanding, including caption and grounded caption. We directly used the instruction examples in the paper Wang et al. (2024). For

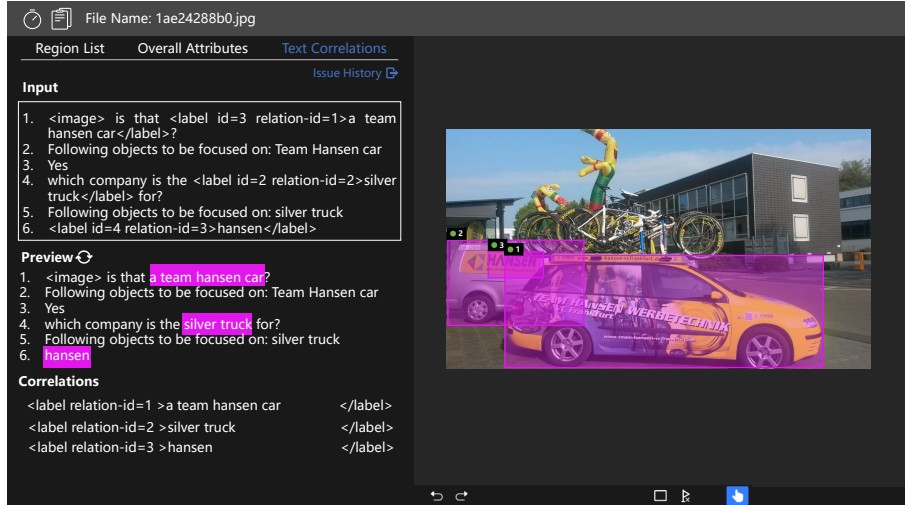

Figure 5: An example UI screenshot showcasing the annotation process for a question-answer pair requiring grounding. After reviewing and confirming the understanding process, the human verifying expert highlights the text based on the knowledge generated by the AI expert and annotates the corresponding bounding box by drawing on the image for each object.

generating this intermediate information at this stage, we perform batch inference using 8 NVIDIA A800 GPUs.

### A.3    VISUALIZATION OF HUMAN EXPERT ANNOTATION

As we have introduced in Sec. 3.3, initial high-quality annotation is important. Therefore, we hire ten human annotation experts to help us. We provide a screenshot example of human expert annotating platform in Fig. 5.

### A.4    DISCUSSION ON DATA SCALING

Several works Chen et al. (2023); Yuan et al. (2024) have highlighted that it's more effective to scale up the annotated data based on the initial high-quality data. Therefore, we follow this insight to employ the human expert for annotation as existing advanced LMMs still fall short in tasks like multi-object visual grounding. For the visual reasoning data scaling up, we provide a brief discussion here. From the perspective of data amount, it's implementable to specifically train an expert modelChen et al. (2023) or our model using the curated data to further annotate a large amount of data. Then, a large expert-level model like ChatGPTOpenAI (2024) or human experts can be used for checking. While for the scene scaling, when considering more scenes, it's possible to include more capabilities into our set and curated related data to support scenes like math, accounting, *etc.* In this paper, we mainly focus on general natural scenes leveraging intrinsic capabilities like REG, caption, and visual grounding, and curate the 334K visual reasoning data. We will follow the discussion to scale up the data and generalize our mechanism to more scenes.

## B    TRAINING DETAILS

### B.1    TRAINING DATA

We list the training data used for the three training stages in Tab. 8, with the data volume and dataset type specified. As we have introduced in Sec. 3.4, we employ the direct supervised fine-tuning (SFT) training strategy to build the mechanism within the Griffon-R. Therefore, we mix the visual reasoning data with general SFT data. While for the overlapping data with the visual reasoning data, we directly remove them.

Table 8: Training data used in each stage. The Lang. represents the language-only instructions, the Inst. represents the vision-language instructions, the Gen. represents the general, the Text. represents text-oriented, and the Perc. represents the perception. Perceptions data include REC, REG, visual grounding and object detection, while VR stands for visual reasoning.

| Stage | Vol. | Type | Training Data |
|---|---|---|---|
| I | 1.2M | Caption | ShareGPT-4V Chen et al. (2023) |
| II | 3.0M | REC | RefCOCO/+Yu et al. (2016), RefCOCOgNagaraja et al. (2016), GRefCOCOHe et al. (2023) |
| | | REG | RefCOCO/+Yu et al. (2016), RefCOCOgNagaraja et al. (2016), GRefCOCOHe et al. (2023), Flickr30K EntitiesPlummer et al. (2017), Visual GenomeKrishna et al. (2017), OspreyYuan et al. (2024) |
| | | DET | Objects365Shao et al. (2019), MSCOCOChen et al. (2015), V3DetWang et al. (2023a), Visual GenomeKrishna et al. (2017) |
| III | 3.9M | Lang. | UltraChatDing et al. (2023), Flan-miniGhosal et al. (2023), OpenOrcaLian et al. (2023), MetaMathQAYu et al. (2023), ShareGPT, MathInstructYue et al. (2023), WizardCoderLuo et al. (2023) |
| | | Inst. | LLaVALiu et al. (2024b), ALLaVAChen et al. (2024), LVIS-Instruct4VWang et al. (2023b) |
| | | Caption | ShareGPT4VChen et al. (2023), TextCapsSidorov et al. (2020) |
| | | Gen. VQA | VQA v2Goyal et al. (2019), GQAHudson & Manning (2019), OK-VQAMarino et al. (2019), A-OKVQASchwenk et al. (2022), SQALu et al. (2022), VizWizBigham et al. (2010) |
| | | Text. VQA | TextVQASingh et al. (2019), OCR-VQAMishra et al. (2019), AI2DKembhavi et al. (2016), SynthdogKim et al. (2022), DVQAKafle et al. (2018), ChartQAMasry et al. (2022), DocVQAMathew et al. (2020), InfoVQAMathew et al. (2022), DeepFormBorchmann et al. (2021), KLCBorchmann et al. (2021), WTQBorchmann et al. (2021), TabFactBorchmann et al. (2021) |
| | | Perc. | RefCOCO/+Yu et al. (2016), RefCOCOgNagaraja et al. (2016), GRefCOCOHe et al. (2023), Flickr30K EntitiesPlummer et al. (2017), Visual GenomeKrishna et al. (2017), OspreyYuan et al. (2024), Objects365Shao et al. (2019), MSCOCOChen et al. (2015), V3DetWang et al. (2023a) |
| | | VR | **Curated Visual Reasoning Data** |

## B.2 TRAINABLE PARAMETER SETTING

In the first stage, we freeze the visual encoder and the LLM and leave the projector trainable. Then, we pretrain the whole model in stage II and further finetune the whole model in stage III.

## C EVALUATION DETAILS

To comprehensively evaluate Griffon-R's capabilities, we conduct a fair comparison with advancing LMMs and visual reasoning methods on visual reasoning benchmarks across both structured scenes like CLEVRJohnson et al. (2017) and natural scenes, encompassing VSRLiu et al. (2023a), GQAHudson & Manning (2019), TallyQAAcharya et al. (2019), and V-Star$_{Spat.}$. These benchmarks mainly require models to comprehend the multi-level information based on the image to infer the final answer. We also evaluate Griffon-R on multimodal benchmarks to demonstrate its comprehensive capabilities. We include benchmarks that contain partial visual reasoning with common sense and knowledge, like MMBenchLiu et al. (2023c), ScienceQALu et al. (2022), SEEDLi et al. (2023a), and LLaVA BenchLiu et al. (2024b), and also TextVQASingh et al. (2019), which focuses on text-scene understanding. Given that strong visual reasoning skills help mitigate model hallucinations, we also assess performance on the POPELi et al. (2023c) benchmark. Also, we validate our design specifically in the ablation studies, including understanding quality indicated by grounding task, mechanism, and data.

## D LLM USAGE

Our use of a Large Language Model (LLM) in this work was limited to providing word-level hints and occasional suggestions for sentence improvement. This is in accordance with the ICLR's Code of Conduct on LLM usage.