# OpenReview forum: "Understand, Think, and Answer: Advancing Visual Reasoning with Large Multimodal Models"
_ICLR.cc/2026/Conference — ICLR 2026 Conference Withdrawn Submission_

### Official Review · Reviewer_EA7o · 2025-10-18

**Soundness:** 3
**Presentation:** 2
**Contribution:** 2
**Rating:** 4
**Confidence:** 4

**Summary:**

This paper proposes a unified visual reasoning mechanism for Large Multimodal Models (LMMs) that follows a human-like "understand-think-answer" process. Unlike existing approaches that either use shortcut learning (directly predicting answers) or rely on external tools/multiple forward passes, the proposed method enables models to: (1) analyze questions and gather relevant visual information using intrinsic capabilities like grounding and captioning, (2) engage in self-prompted reasoning based on gathered cues, and (3) generate final answers—all in a single forward pass. The authors introduce a semi-automatic expert-supervised data engine to curate 334K visual reasoning samples and train Griffon-R, which achieves strong performance on visual reasoning benchmarks (VSR: 70.9%, CLEVR: 63.7%) and general multimodal benchmarks (MMBench: 79.0%, ScienceQA: 87.0%).

**Strengths:**

Well-motivated approach: The "understand-think-answer" paradigm is intuitive and addresses real limitations of current LMMs in compositional reasoning. The motivation is clearly articulated with concrete examples (Figure 1).

Efficiency advantage: Completing all reasoning steps in a single forward pass without external tools is more efficient than toolkit-based methods. Table 4 shows 13x speedup over SEAL while maintaining competitive accuracy.

Strong empirical results: Griffon-R achieves state-of-the-art or competitive performance across multiple benchmarks, particularly excelling on CLEVR (63.7% vs. 55.8% for the second-best) and showing improvements over both standard LMMs and enhanced reasoning methods.

Comprehensive evaluation: The paper evaluates on both specialized visual reasoning benchmarks and general multimodal tasks, demonstrating that the method doesn't sacrifice general capabilities for reasoning performance.

Practical data generation pipeline: The semi-automatic annotation approach combining AI expert (Qwen2-VL-72B) and human curation is a pragmatic contribution, though it needs more detail.

**Weaknesses:**

1. Limited Technical Novelty

The core contribution appears to be more about structured data annotation than a novel reasoning mechanism:

The "understanding" step simply prompts the model to use existing capabilities (grounding, captioning, OCR) it already possesses

The "thinking" step is self-prompting based on context, which is well-explored in CoT literature

The main innovation is in training data format rather than architecture or fundamental mechanisms

2. Insufficient Detail on Data Annotation Process

Critical aspects of the data engine are under-specified:

What are the exact prompts used to instruct the AI expert? (Referenced Appendix A not provided)

What percentage of annotations required human correction or removal?

How much human effort was needed? This is crucial for assessing scalability

What quality control metrics were used?

The 334K dataset seems relatively small for modern LMM training—how does scale affect performance?

3. Weak Mechanism Validation

The ablation studies don't convincingly validate the three-stage design:

Table 4 compares against only one baseline (SEAL) on one benchmark (V-Star) with modest improvement (+1.3 points)

No ablation showing the necessity of all three stages (understand/think/answer). What if understanding+answer suffice? Or just think+answer?

Table 5 shows annotations help but doesn't decompose which aspects (understanding annotations vs. thinking annotations) matter most

4. Incomplete Analysis of Understanding Quality

Table 3 uses RefCOCO as a proxy for understanding quality, but this only measures visual grounding, not comprehensive understanding

No analysis of: How often does the understanding stage gather correct/relevant information? How do understanding errors propagate?

Missing failure case analysis—when does the mechanism break down?

No error breakdown by reasoning type or task complexity

5. Unfair or Incomplete Comparisons

Models compared have different parameter counts (7B, 9B, 13B, 17B) making direct comparison difficult

Some baselines are older (e.g., LLaVA-1.5 from 2023); missing comparisons with more recent strong models

Table 1 shows missing results for several method-benchmark combinations (many "-" entries)

No comparison with recent visual reasoning methods like GPT-4V with structured prompting

6. Scalability and Generalization Concerns

Requires expensive, carefully curated three-stage annotations—can this scale to millions of samples?

Tested only on Griffon architecture with Gemma9B—does it generalize to other LMM architectures?

Can the model learn to generate intermediate steps without explicit supervision (e.g., via RL or distillation)?

**Questions:**

Can you provide complete ablations removing each stage independently to validate the three-stage design?

What percentage of the 334K annotations required human intervention, and what was the inter-annotator agreement?

Does this approach generalize to other LMM architectures beyond Griffon? Have you tested on LLaVA or Qwen-VL bases?

Can you provide detailed error analysis showing when understanding/thinking/answering fails?

What are the exact inference-time prompts for understanding and thinking stages?

Have you explored learning to generate intermediate steps without explicit annotation (e.g., through reinforcement learning or self-improvement)?

How does performance scale with the amount of annotated training data?

---

### Official Review · Reviewer_ngzi · 2025-10-29

**Soundness:** 3
**Presentation:** 2
**Contribution:** 3
**Rating:** 4
**Confidence:** 4

**Summary:**

This paper introduces Griffon-R, a large multimodal model for visual reasoning based on a unified Understand–Think–Answer (UTA) process.
Instead of relying on external tools or multi-step reasoning, Griffon-R performs end-to-end reasoning in a single forward pass, mimicking how humans perceive, think, and respond.
Trained on a 334K expert-curated dataset, it achieves strong results on benchmarks like VSR, CLEVR, and MMBench, outperforming prior LMMs while being much faster.
The work offers a concise, efficient, and interpretable approach to multimodal reasoning.

**Strengths:**

1. The work pushes forward end-to-end compositional reasoning, a core obstacle in vision-language integration. It establishes a strong empirical and conceptual foundation for future intrinsically capable multimodal reasoning models, potentially influencing both academic and applied LMM research.
2. The paper provides a high-quality dataset (334K samples) created through a semi-automatic expert-supervised process — a valuable contribution for the community.

**Weaknesses:**

1. The paper lacks a formal theoretical analysis and ablation experiments quantifying the specific contribution of each UTA phase.
2. Generalization to out-of-domain or noisy visual data remains unclear.
3. Many references are incorrectly formatted or missing spaces between citations and text, which affects readability and professionalism.

**Questions:**

1. How is the “self-prompt” mechanism implemented internally? Is it learned implicitly through training on annotated reasoning sequences, or guided by fixed templates during inference?
2. Will the curated 334K dataset or annotation engine be publicly released? This is crucial for reproducibility and comparison.
3. The paper emphasizes traceable reasoning. Could the authors quantify this with metrics like faithfulness or rationalization alignment?

---

### Official Review · Reviewer_yews · 2025-10-30

**Soundness:** 2
**Presentation:** 2
**Contribution:** 1
**Rating:** 2
**Confidence:** 4

**Summary:**

This work proposes a unified visual reasoning CoT format and demonstrates a data curation pipeline. Training VLM with the collected data results in enhanced performances on multiple multimodal understanding and reasoning benchmarks.

**Strengths:**

The paper provides enough details to help the readers understand the full story. The experimental results cover multiple domains.

**Weaknesses:**

1. No novel methodology or insights proposed: the unified framework features no significant difference compared to widely-used CoTs for VLMs. The motivation of unifying visual reasoning format for different tasks is unclear, as special tasks (math, coding, ...) need special CoT formats, and advanced LLMs can discover new formats via RL.
2. The compared models and benchmarks are severely outdated for a work in 2025. For example,they used Qwen-2-vl for data annotation, while Qwen2.5-vl is released in 2025.02. Very few commonly-used VLM benchmarks after 2024 is evaluated.
3. The paper writing is not in good style: the citation formats have no parentheses; appendix is in supplementary instead of the main pdf.

**Questions:**

Most details are clear, but the paper may benefit from updating the models and benchmarks.

---

### Official Review · Reviewer_wtzv · 2025-10-31

**Soundness:** 1
**Presentation:** 2
**Contribution:** 1
**Rating:** 2
**Confidence:** 4

**Summary:**

The authors fine-tune a Gemma model to get their proposed Griffon-R model, using data that has been labeled by experts/AI. They find the fine-tuned model performs well on data similar to the labeled data. They do this using a proposed prompt engineering approach, which they call an "understand-think-answer" process.

**Strengths:**

- The results in Table 2 were strong.

**Weaknesses:**

- The "Understand-Think-Answer" approach is a relatively common prompt engineering approach and is not novel nor a significant research contribution.
- The authors add a new dataset, test their fine-tuned model on data similar to their new dataset, and report strong results. This likely results in overfitting to their fine-tuned dataset and forgetting of old information, which should be tested for.
- The benchmark comparisons made are unfair as they compare a model with more data that has been labeled then the existing models. They should compare the proposed model with other models having seen the same data.
- Incorporating AI into the annotation results in noisier labeled data.
- The authors abstain from making it clear the proposed griffon model is a fine-tuned variant of Gemini.

**Questions:**

N/A

---

### Note · Authors · 2026-01-06

I have read and agree with the venue's withdrawal policy on behalf of myself and my co-authors.